# Depletion Interactions at Interfaces Induced by Ferromagnetic Colloidal Polymers

**DOI:** 10.3390/polym16060820

**Published:** 2024-03-15

**Authors:** Joan Josep Cerdà, Josep Batle, Carles Bona-Casas, Joan Massó, Tomàs Sintes

**Affiliations:** 1Departament de Física UIB i Institut d’Aplicacions Computacionals de Codi Comunitari (IAC3), Campus UIB, E-07122 Palma de Mallorca, Spain; 2Centre de Recerca Independent, E-07420 Sa Pobla, Spain; 3Instituto de Física Interdisciplinar y Sistemas Complejos, IFISC (CSIC-UIB), Universitat de les Illes Balears, E-07122 Palma de Mallorca, Spain

**Keywords:** colloidal polymers, magnetism, numerical simulations

## Abstract

The pair-interaction force profiles for two non-magnetic colloids immersed in a suspension of ferromagnetic colloidal polymers are investigated via Langevin simulations. A quasi-two-dimensional approach is taken to study the interface case and a range of colloidal size ratios (non-magnetic:magnetic) from 6:1 up to 20:1 have been considered in this work. Simulations show that when compared with non-magnetic suspensions, the magnetic polymers strongly modify the depletion force profiles leading to strongly oscillatory behavior. Larger polymer densities and size ratios increase the range of the depletion forces, and in general, also their strength; the force barrier peaks at short distances show more complex behavior. As the length of the ferromagnetic polymers increases, the force profiles become more regular, and stable points with their corresponding attraction basins develop. The number of stable points and the distance at which they occur can be tuned through the modification of the field strength *H* and the angle θ formed by the field and the imaginary axis joining the centers of the two non-magnetic colloids. When not constrained, the net forces acting on the two colloids tend to align them with the field till $\theta =0^{\circ }$. At this angle, the force profiles turn out to be purely attractive, and therefore, these systems could be used as a funneling tool to form long linear arrays of non-magnetic particles. Torsional forces peak at $\theta =45^{\circ }$ and have minimums at $\theta =0^{\circ }$ as well as $\theta =90^{\circ }$ which is an unstable orientation as slight deviations will evolve towards $\theta \to 0^{\circ }$. Nonetheless, results suggest that the $\theta =90^{\circ }$ orientation could be easily stabilized in several ways. In such a case, the stable points that the radial force profiles exhibit for this orthogonal orientation to the field could be used to control the distance between the two large colloids: their position and number can be controlled via *H*. Therefore, suspensions made of ferromagnetic colloidal polymers can be also useful in the creation of magnetic colloidal tweezers or ratchets. A qualitative explanation of all the observed phenomena can be provided in terms of how the geometrical constraints and the external field modify the conformations of the ferromagnetic polymers near the two large particles, and in turn, how both factors combine to create unbalanced Kelvin forces that oscillate in strength with the distance between the two non-magnetic colloids.

## 1. Introduction

Depletion interactions are ubiquitous in nature and can be found at the root of many biological interactions as well as being involved in many industrial applications [1,2]. These interactions stem from the attempt to maximize the entropy of mixtures of different components. The understanding of these depletion interactions has been a long-run field of research in physics: for a review see the book of Lekerkerker-Tuinier [3]. First observations and the study of the phenomena date back to the end of the 18th century and early 19th century: some of the first reports on the field arose from the attempted mixing of gelatin and starch, the enhanced aggregation of the red blood cells [4], or the creaming of particles in latex [5].

Before the advent of experimental direct measurement techniques, the only way to obtain accurate experimental results was via the measurement of the structure factors of colloidal dispersions using available scattering methods. Once structure factors were obtained, pair correlation functions could be computed using Fourier transformations. In turn, these pair correlation functions were related to the potentials associated with the depletion forces via statistical mechanics. Nonetheless, the previous methodology was not exempt from problems, as results were sensitive to the selected closure relation used to compute the potentials from the pair correlation functions. Current direct measurement techniques provide a more straight path to determine these forces, although their experimental characterization continues to be difficult as their usual strength is just of few kT’s. The most common direct experimental techniques available are internal reflection microscopy (TIRM) [6,7,8,9], optical tweezers (OT) [10], scanning force microscopy (SFM) [11], surface force apparatus SFA [12], and the magnetic chaining technique [13]. For a review related to experimental research on polymer depletion-induced forces see for instance Kleshchanok et al. [14] and Tuinier et al. [15].

Another approach to the problem of characterizing depletion interactions is the development of theoretical frameworks [14,15]. In 1954, S. Asakura and F. Oosawa [16] published their seminal work in which they explained the depletion phenomena making use of asymmetrical osmotic pressure arguments. Since then, a corpus of theoretical work has been devoted to the study of depletion interactions for different types of mixtures and constraints using different theoretical approaches. Thus, for instance, the application of scaling theories [17,18] and mean-field methods [18,19,20] to the problem was initiated in the 1970s. Further improvements for the polymer-induced depletions in a colloidal suspension were introduced in the 1980s by De Hek and Vrij [21], Gast et al. [22], and in the 1990s by Taniguchi [23], Eisenriegler [24], and Lekkerkerker [25] who developed the osmotic equilibrium theory. The calculation of the depletion potentials has been approached also in the last two decades via density functional theories [26,27,28,29], and several refining theories like liquid state theories [30] and Gaussian core models for interacting polymer chains [31,32,33]. The progress of these theoretical frameworks has run in parallel with the incorporation into the theories of different features like the shape [34,35,36,37,38,39,40], charge [41,42,43,44,45,46,47,48], size [9], polydispersity [15,49,50,51], and softness [19,52] of the particles and depletants. Theories have also been further developed to cover more complex systems like those containing interacting depletants [53,54,55,56], multiple depletant types [57], surfactants [57,58,59], polymer brushes [60], or that exhibit multi-body effects [61,62,63,64]. Theories have also addressed the case of active colloids immersed in a bath of passive depletants [65], nanobubbles [66], nanoparticles [67,68,69,70,71], microgels [72], and bacterial systems [73]. One should also mention the theoretical efforts intended to characterize the depletion phenomena in confined domains [74,75], as well as near dielectric discontinuities [76] and in shear conditions [77]. For a review of the research about the topic in both bulk and confined domains see for instance Trokhymchuk et al. [74].

In addition to theoretical developments, another fruitful approach to the study of depletion interactions is the use of numerical simulations. Simulations allow to study systems and regimes that can be difficult to model using theoretical frameworks. Simulations can also help to test the accuracy of the different theories: for instance, failures of early theories in the ‘protein limit’ (relative large polymer chains to colloids size) were exposed by Monte Carlo numerical studies [31,78,79,80,81]. These failures were then subsequently addressed by the improved theoretical approaches of Van der Schoot [82] and those based on the polymer reference interaction site model (PRISM) [83]. Numerical simulations have also been used in determining the depletion potentials for many different types of systems, shapes and constraints [46,56,84,85,86,87,88,89,90,91,92,93,94,95,96,97,98,99,100,101], including the study of very high particle concentrations [102], active particles [103,104,105,106,107], and biological systems [108]. Simulations have also helped to study the phase diagram [109,110,111,112] and the self-assembling [113,114,115,116] in systems ruled by these forces.

In this work, we propose to study and characterize the depletion interactions of a pair of non-magnetic colloidal particles at an interface surrounded by a solution of ferromagnetic polymers. These magnetic polymers or filaments are equivalent at the supramolecular scale of the chemical magnetic polymers: they are formed by sequentially linking magnetic colloidal particles to form a chain of colloids (see Ref. [117] and cites therein). The depletion interactions induced by these magnetic colloidal polymers have never been addressed before and, as we will show in this work, they exhibit very interesting properties.

The manuscript is organized as follows. In the next section, the numerical model and the details of the simulations are described. The main findings and discussions are provided in Section 3, and the concluding remarks are provided in Section 4.

## 2. Numerical Method

Our model mimics a quasi-two-dimensional system in which the centers of two large non-magnetic soft colloidal particles of radius $R_{e}$ are placed in a plane that also contains a solution of ferromagnetic colloidal polymers, see Figure 1. We will refer to the ferromagnetic colloidal particles forming part of these colloidal polymers as monomers. Monomers are assumed to be monodisperse in size and have a diameter $\sigma_{e}$. Henceforth, subindex *e* denotes the values of the physical quantities in non-reduced units, and the lack of subindex implies they are expressed in reduced units. For instance, in our simulations, length scales are measured in units of the monomers $\sigma_{e}$. Therefore, a measured length in experiment $l_{e}$ corresponds to a reduced value $l=l_{e}/\sigma_{e}$. Consequently, in all our simulations the diameter in reduced units of monomers is set to σ=1, and the radius of the two large non-magnetic colloidal particles in reduced units is $R=R_{e}/\sigma_{e}$.

As our system mimics a first approach to an interface, all the particles in the system have their center of mass confined in the same plane. The value of *L* has been chosen in the range L∈[50,90] depending on the size of the two large non-magnetic colloidal particles, the length of the ferromagnetic colloidal polymers, and the strength of the external field.

The surface density of monomers in our quasi-two dimensional systems, ϕ is computed according to the effective area available to them as
(1)$$\phi =\frac{N_{c}N_{p}\pi {\left(\sigma /2\right)}^{2}}{L^{2}-2\pi R^{2}}$$
where $N_{c}$ is the number of ferromagnetic colloidal particles (monomers) forming a polymer chain, and $N_{p}$ is the total number of ferromagnetic polymers present in the system. In Equation (Equation 1) we subtract from the total area $L^{2}$ the area occupied by the two large non-magnetic particles $2\pi R^{2}$ in order to obtain an estimate of the effective area available to the polymer chains which is the observable directly related to the entropy of those chains.In our study, we have tested surface densities in the range ϕ∈(0.05,0.35).

Colloidal polymers are modeled as bead-spring chains. The steric interaction among all the monomers in the system is modeled using a Weeks–Chandler–Anderson (WCA) potential (short-ranged shifted Lennard–Jones potential truncated at its minimum) [118]: two monomer particles *i* and *j* interact via
(2)$$V_{tsLJ}\left(r,\sigma ,\epsilon ,r_{cut}\right)=\left\{\begin{array}{cc} U_{LJ}\left(r\right)-U_{LJ}\left(r_{\mathrm{cut}}\right), & \mathrm{for}\;r<r_{\mathrm{cut}}\\ 0, & \mathrm{for}\;r\geq r_{\mathrm{cut}} \end{array}\right.,$$
where $U_{LJ}\left(r\right)=4\epsilon \left[{\left(\sigma /r\right)}^{12}-{\left(\sigma /r\right)}^{6}\right]$, *r* is the distance between the centers of particles *i* and *j*, i.e., $r=|r_{i}-r_{j}|$, and $r_{\mathrm{cut}}=2^{1/6}\sigma$ is the cut-off radius. The LJ energy parameter ϵ in Equation (Equation 2) is given in units of the experimental well depth $\epsilon_{e}$ between two monomers, and any energy *U* will be also referred to $\epsilon_{e}$, i.e., $U=U_{e}/\epsilon_{e}$. We choose the Boltzmann constant to be k=1 in reduced units, and therefore, the reduced temperature in our simulations can be obtained as $T=k_{e}T_{e}/\epsilon_{e}$. In the previous formula $k_{e}T_{e}$ is the product of the Boltzmann constant and the experimental temperature expressed in the same system of units as $\epsilon_{e}$. The steric interaction among monomers and the two large non-magnetic colloids is modeled using also a WCA potential as in Equation (Equation 2) but replacing σ by R+σ/2. We do not set any direct interaction among the two large particles: in this way when we measure the force acting on them we directly measure only the interaction arising from the surrounding ferromagnetic suspension. This approach is expected to provide the sought force profiles as far as the two non-magnetic particles do not overlap, i.e., d≥0 (see Figure 1).

In order to connect the monomers that form a colloidal polymer, we use a finite extension non-linear elastic (FENE) linking model [119],
(3)$$U_{s}\left(r\right)=-\frac{1}{2}K_{s}\Delta r_{max}^{2}\ln\left[1-{\left(\frac{r-r_{0}}{\Delta r_{max}}\right)}^{2}\right],$$
where the constants of the potential are set to $r_{0}=\sigma$, Δr=1.5σ and $K_{s}=15$.

All monomers are assumed to bear a point dipole of modulus $\mu_{e}$ located at their center. While particles are restricted to move on the x−y plane, their dipoles are allowed to rotate in all 3D directions: this is known as a quasi-two-dimensional approach (Q2D). The modulus can be expressed in the reduced unit system as: $\mu^{2}=\mu_{e}^{2}/\left(4\pi \mu_{o,e}\sigma_{e}^{3}\epsilon_{e}\right)$, where $\mu_{o,e}$ is the vacuum permeability, and the SI electromagnetic units have been assumed to express the experimental magnetic energy associated with a system of two magnetic monomers at close contact with their dipoles aligned nose-to-tail. Therefore, in our reduced unit system, two monomers are assumed to interact pair-wise as point dipoles according to the potential,
(4)$$U_{dip}\left(r_{ij}\right)=\frac{\mu_{i}\cdot \mu_{j}}{|r_{ij}{|}^{3}}-\frac{3\left[\mu_{i}\cdot r_{ij}\right]\left[\mu_{j}\cdot r_{ij}\right]}{|r_{ij}{|}^{5}},$$
where $r_{ij}=r_{i}-r_{j}$ is the displacement vector between particles *i* and *j*. Reasonable values of μ=|μ| depend, in general, on the composition and size of the colloidal particles. In this study, we assume for all magnetic particles $\mu^{2}=5$. As a reference, colloidal particles found in common commercial ferrofluids usually do not exceed values of μ∼10, except in the case of cobalt nanoparticles that are substantially higher. The long-range dipole–dipole interactions in this Q2D geometry are calculated by combining $dP^{3}M$ [120] and DLC [121] methods. The DLC correction must be applied to discount the effect of the excess of infinite replicas added along the third direction by the $dP^{3}M$ method that assumes 3D periodic boundary conditions. The combined use of $dP^{3}M$ and DLC allows a much faster calculation of the dipolar long-range interactions than the traditional two-dimensional dipolar Ewald summations [122] adapted to slit geometries. The level of accuracy of the algorithm for computing dipolar forces and torques is set in our simulations to δ∼$10^{-3}$.

Simulations in which a homogeneous external magnetic field of strength *H* is applied have an additional potential acting on monomers: $U_{H}=-\mu \cdot H$ that leads to an extra contribution to the torque acting on the dipoles: τ=μ×H. In this study, we have focused on fields in the range |H|∈[0,1], which correspond to values of the Langevin Parameter $\alpha \equiv \mu \cdot H/\left(kT\right)\in \left[0,5^{1/2}\right]$.

The numerical simulations are performed using Langevin dynamics. The coordinates of the monomers are moved according to the translational and rotational Langevin equations of motion that for a given monomer *i* are [123]:(5)$$\begin{array}{c} M_{i}\frac{dv_{i}}{dt}=F_{i}-\Gamma_{T}v_{i}+\xi_{i}^{T} \end{array}$$(6)$$\begin{array}{c} I_{i}\cdot \frac{d\omega_{i}}{dt}=\tau_{i}-\Gamma_{R}\omega_{i}+\xi_{i}^{R} \end{array}$$
where $F_{i}$, and $\tau_{i}$ are, respectively, the total force and torque acting on monomer *i* due to all other particles. As we pursue a Q2D approach, and particles are lying on the x-y plane in our simulations, the z-spatial coordinates are kept constant by not integrating them. In our simulations, periodic boundary conditions are used.

The magnetic component of the torque, $\tau_{i}^{\left(mag\right)}$, can be computed as
(7)$$\tau_{i}^{\left(mag\right)}=-\mu_{i}\times \nabla_{\mu_{i}}\left(U_{\mathrm{dip}}+U_{H}\right).$$
$M_{i}$ and $I_{i}$ are the mass and inertia tensors of the bead, and $\Gamma_{T}$ and $\Gamma_{R}$ are the translational and rotational friction constants. $\xi_{i}^{T}$ and $\xi_{i}^{R}$ are Gaussian random forces and torques, each of zero mean and satisfying the usual fluctuation-dissipation relations (Greek letters α and β represent components *x*, *y* or *z*):(8)$$\begin{array}{ccc} \langle \xi_{i\alpha }^{T}\left(t\right)\xi_{j\beta }^{T}\left(t^{\prime }\right)\rangle & = & 2kT\Gamma_{T}\delta_{ij}\delta_{\alpha \beta }\delta \left(t-t^{\prime }\right), \end{array}$$(9)$$\begin{array}{ccc} \langle \xi_{i\alpha }^{R}\left(t\right)\xi_{j\beta }^{R}\left(t^{\prime }\right)\rangle & = & 2kT\Gamma_{R}\delta_{ij}\delta_{\alpha \beta }\delta \left(t-t^{\prime }\right). \end{array}$$

In the simulations, $t=t_{e}\sqrt{\epsilon_{e}/\left(m_{e}\sigma_{e}^{2}\right)}$, where $m_{e}$ is the real mass of the colloids forming part of the magnetic colloidal polymers; the correspondence of forces and torques in reduced units with experimental values are $F=F_{e}\sigma_{e}/\epsilon_{e}$, and $\tau =\tau_{e}/\epsilon_{e}$, respectively. For simplicity, the mass of all particles has been set to m=1 in reduced units. For the translation motion, we use the friction constant $\Gamma_{T}=1$ in reduced units. For equilibrium simulations, the values of the mass, the inertia tensor, as well as friction constants $\Gamma_{T}$, and $\Gamma_{R}$ are irrelevant because the same equilibrium state is reached independently of their value. Only the dynamics to attain such an equilibrium state may show differences. The time step is set to $\delta t=5\times 10^{-3}$ and temperature to T=1.0 in reduced units. The simulations have been run using the package ESPResSo [124], version 3.3.1.

In our study, a typical run starts by placing the two non-magnetic colloidal particles at close contact, d=0, onto the x-y plane (see Figure 1): the non-magnetic particle labeled as 1 in Figure 1 is located at (L/2−R,L/2), while the particle labeled as 2 is placed at (L/2+R,L/2). The position of particles 1 and 2 will be held fixed during the whole equilibration and measurement stages for a given distance *d*: this is easily accomplished by not integrating their positions in the Langevin equations. Next, while keeping magnetic interactions turned off, the colloidal polymers are introduced randomly in the plane using partial self-avoiding random walks. Polymers are prevented from overlapping with the two non-magnetic colloidal particles. The system is then equilibrated till the minimum distance among monomers is larger than 0.88σ, and the minimum distance between monomers and the non-magnetic colloidal particles is also larger than R+σ/2. At this point, the magnetic interactions among the monomers are activated and an equilibration of $2\times 10^{6}$ time steps is performed. Subsequently, if H>0 the external field is turned on and the system is further equilibrated another $1.5\times 10^{6}$ time steps. Five runs using different initial random seeds have been performed in order to obtain the averaged force profiles shown in this work.

Once the system has been created, a cycle of measures to obtain the forces acting on the two non-magnetic particles as a function of the distance *d* between their centers starts. The increase in the distance between the two particles from *d* to d+Δd is conducted by performing a short sequence of small displacements of their positions, (±0.001σ,0,0), followed by a 100 time step equilibration after each change until the new relative position between the two non-magnetic colloids is attained. This process is followed by a $2\times 10^{6}$ time step equilibration holding fixed the position of the non-magnetic colloids. Next, forces acting onto the two non-magnetic particles (always keeping fixed their position) are recorded over a period of $30\times 10^{6}$ time steps at intervals of 100 time steps: this is conducted by computing the force acting on both of them due to their interaction with the monomers of the ferromagnetic polymers. Once the sequence of measurements is completed for a given distance, a new iteration of the cycle is performed until the desired final distance d/σ=7 between the two non-magnetic colloids is obtained.

We expect our simulations to give the correct equilibrium properties in the canonical ensemble even though no hydrodynamic effects have been included in the model. This claim is based on the fact that as far as the generalized equation of Smoluchowski with hydrodynamics (GESH) can be used as a valid description of the system, its stationary solution is the density probability function of the canonical ensemble [125]. The GESH equation is expected to hold on Brownian time and length scales where both particles and solvent are quasi-inertia-free. In the scale we are performing our measurements the inertial forces must be very small compared to the forces arising from the total potential, the Brownian forces, and the forces of friction with the solvent. We assume that in our system previous conditions hold.

## 3. Results and Discussion

Henceforth, for the sake of simplicity, we will refer to the axis joining the center of mass of the two larger non-magnetic particles as the joining axis. The angle formed between this joining axis and the direction along which the external field is applied will be labeled as θ. In our setup, see Figure 1, a value of $\theta =0^{\circ }$ implies that the field is parallel to the x-axis, while $\theta =90^{\circ }$ implies that the field is parallel to the y-axis.

In order to measure the depletion force onto the two non-magnetic colloids, we have monitored the forces induced by the solution of magnetic colloidal polymers on both particles: $f_{1}=\left(f_{1}^{\left(x\right)},\;f_{1}^{\left(y\right)}\right)$ and $f_{2}=\left(f_{2}^{\left(x\right)},\;f_{2}^{\left(y\right)}\right)$, respectively. As expected by the symmetry of the problem, our simulations show that on average $f_{2}=-f_{1}$ in all cases. In general, for an arbitrary angle θ the component of the force perpendicular to the joining axis is non-zero, $\langle f^{\left(y\right)}\rangle \neq 0$. Nonetheless, when the external field is zero or if an external field is applied but *θ* = 0° or *θ* = 90°, then on average this perpendicular component of the force is zero. In our setup, this means that in those three cases $\langle f_{1}^{\left(y\right)}\rangle =\langle f_{2}^{\left(y\right)}\rangle =0$. In such cases, the system can then be characterized by simply monitoring the depletion force $f_{2}^{\left(x\right)}$ as a function of the closest distance between the two colloidal particles $d=x_{2}-x_{1}-2R$, where, as shown in Figure 1, $x_{1}$ and $x_{2}$ are the positions along the x-axis of the centers of the two large non-magnetic colloids. Figure 1 also depicts a typical depletion force profile for $f_{2}^{\left(x\right)}$: as shown, $f_{2}^{\left(x\right)}>0$ implies a repulsive depletion force between the two non-magnetic particles, and $f_{2}^{\left(x\right)}<0$ implies the existence of an attractive depletion force. For those distances $d_{s}$ at which $f_{2}^{\left(x\right)}\left(d_{s}\right)=0$ and ${\left.\frac{\partial f_{2}^{\left(x\right)}}{\partial d}\right|}_{d=d_{s}}<0$ the two large particles will tend to not modify their distance, as the system is in a local minimum energy or potential well, and henceforth, we will refer to these cases as stable positions. On the other hand, for those distances $d_{u}$ such that $f_{2}^{\left(x\right)}\left(d_{u}\right)=0$ and ${\left.\frac{\partial f_{2}^{\left(x\right)}}{\partial d}\right|}_{d=d_{u}}>0$ the system is in a local maximum of the potential, and we will refer to those situations as unstable positions in the system: the slightest perturbation will lead the two large particles to either become closer or further separate. For a given stable point located at $d=d_{s}$ its basis of attraction, which will correspond to a well in the potential, can be defined as the range of d-values between two consecutive unstable positions $d_{u}^{\left(1\right)}$ and $d_{u}^{\left(2\right)}$ such that $d_{s}\in \left(d_{u}^{\left(1\right)},d_{u}^{\left(2\right)}\right)$.

### 3.1. Force Profiles for H=0


We will start the analysis of the depletion forces by studying two non-magnetic colloidal particles in a bath of non-linked monomers: $N_{c}=1$ case. Figure 2 compares the depletion force profiles $f_{2}\left(d\right)$ for a suspension of non-magnetic (black dashed lines with empty symbols) and magnetic (solid lines with filled symbols) particles when the external magnetic field is zero. Results show that the magnetic case strongly departs from the force profile corresponding to a non-magnetic suspension. Differences can be attributed to the formation of small particle aggregates due to the magnetic interaction among colloids, which is a well-known behavior observed in ferrofluids [126]. This magnetic aggregation phenomenon also explains why the cases $N_{c}=10$ non-magnetic and $N_{c}=1$ magnetic have quite similar force profiles, as in both the entropic effects leading to depletion forces are expected to be induced by depletant aggregates of comparable size.

In all the cases studied, Figure 2 also shows that in the close contact region (d/σ<1) the attractive force between colloids weakens when a solution of polymers or non-linked magnetic monomers is considered. It is also remarkable that for $N_{c}=20$ and $\mu^{2}=5$ a set of stable points emerges in the region d/σ∈(2,4), where oscillations are largely amplified and the period is shortened leading to narrower attraction basins with larger restoring forces towards the equilibrium positions.

The oscillations with a larger amplitude and shorter wavelength observed in Figure 2 for ferromagnetic polymers in the range d/σ∈(2,4) can be related to the fact that ferromagnetic polymers can pack more densely than their non-magnetic polymer counterparts in the region gap between the two large non-magnetic colloids. In the case of the ferromagnetic polymers, their dipoles tend to orientate in a nose-to-tail configuration which leads, on average, to more stretched polymer conformations. For short ferromagnetic chains, the local persistence length is expected to increase with $N_{c}$. As ferromagnetic colloids can, therefore, pack more densely, the required change in the gap distance Δd determines if another polymer can be easily accommodated or not into the gap shortenings. This is expected to lead to oscillations of shorter wavelengths as the length of the oscillation in the depletion force is expected to be influenced by Δd. In turn, larger amplitude oscillations in the depletion force profiles are expected to occur as the density gradients created by a slightly varying distance *d* will be larger. It should be mentioned that oscillations in the osmotic pressure with wall separation related to the density have been also observed in systems of colloidal particles immersed in a sea of charged nanoparticles [96].

It should be noticed that the depletion force profiles in these systems have more complex details than traditional Asakura–Oosawa potentials: one must take into account the fact that the typical ratio of polymer size to the non-magnetic colloidal particles is not negligible and, in addition, all particles interact through soft core potentials instead of hard core potentials.

The position of the stable points and the range of the attractive basins could be somehow modified by changing the parameters of the system. Nonetheless, magnetic systems have an additional advantage over non-magnetic ones, as they are responsive to external magnetic fields, which are known to boost the aggregation of magnetic particles and chains along the field direction. Thus, one can explore the possibility of externally tuning the steric interaction and modifying the depletion force profiles by modifying both the strength and orientation of an external magnetic field.

### 3.2. Force Profiles for H>0

The analytical prediction of the force profiles for the systems under study when an external field is applied is highly non-trivial. An external field not only modifies the properties and apparent size of the polymer chains in suspension but also introduces an extra contribution to the force profile due to Kelvin forces. This extra contribution arises from the magnetic mismatch between the non-magnetic particles and the magnetic suspension. A clear example of them in soft matter occurs in the magnetophoresis of spheres of weakly magnetic materials, where the sum of all these forces onto single particles immersed in a medium can be resumed in the following expression [127]:(10)$$F_{mag}=\frac{\chi_{p}-\chi_{m}}{2\mu_{o}}V\nabla {\left(B\right)}^{2},$$
where, $\chi_{p}$ and $\chi_{m}$ represent the volume magnetic susceptibilities of the particle and the medium, respectively. *V* is the volume of the particle, and $B=\mu_{0}H+M$, being ***M*** the magnetization and $\mu_{o}$ the permeability of the vacuum. Another well-known example is inverse ferrofluids [128,129]: these systems can be modeled, in a first approach, as if the magnetic suspension was not present and the non-magnetic particles of radius *R* were instead bearing an induced magnetic dipole $\mu =-4\pi \beta R^{3}H$, where $\beta =\left(\mu_{r}-1\right)/\left(2\mu_{r}+1\right)$ characterizes the effective permeability of the magnetic suspension, and $\mu_{r}$ represents the relative permeability of the fluid. This alternative image used to predict the behavior of inverse ferrofluids will be very useful when discussing our results, although the previous Equation (Equation 10) cannot be applied quantitatively. This limitation is due to the fact that the colloidal particles forming part of the polymeric chains are highly correlated: they are linked forming chains, and therefore, their density is perturbed by the existence of the two non-magnetic particles. Thus, we cannot assume a homogeneous bath of magnetic particles as the use of Equation (Equation 10) implies. In addition, the aggregates in suspension have a size that cannot be disregarded with respect to the size of the two non-magnetic particles which would be another condition to be fulfilled for Equation (Equation 10) to hold.

It is worth remarking that in our systems Kelvin forces are zero if H=0: in that case, symmetry arguments imply that magnetization of the suspension at any point of it must be zero on average, and therefore, no mismatch in the magnetic susceptibilities of the suspension and the non-magnetic particles is expected. On the other hand, if an external field ***H*** is applied, the symmetry of the system is broken and, a non-zero magnetization of the suspension is expected: in that case, in the region near the surfaces of the non-magnetic particles ∇B≠0, and therefore, Kelvin forces are expected to emerge.

In the first step, we focus on the two simplest cases: $\theta =0^{\circ }$ (in our setup H=(H,0)), and $\theta =90^{\circ }$ (in our setup H=(0,H)). Figure 3a–d depicts the force profiles obtained for suspensions of ferromagnetic polymers of length $N_{c}=\;1,\;10,\;20$ when $\theta =90^{\circ }$ at several field strengths H=0.02,0.1,0.5,1. When comparing the force profiles with and without an external field, compare Figure 3 with Figure 2, it is noticeable that an increase in *H* leads to more regular oscillating force profiles with larger repulsive barriers to overcome, although the attractive basin for d/σ<1 never disappears. In addition, an external field applied at $\theta =90^{\circ }$ adds a repulsive contribution to the force profiles, turning parts of the force profile that were attractive into repulsive. This progressive shifting of the force profile induced by the external field allows the emergence of a larger set of stable points in the force profiles for moderate values of *H*. A comparison of Figure 3b–d shows that by further increasing *H* the force profiles do not show any substantial modification: a saturating force profile is reached. The field threshold for saturation depends on $N_{c}$: larger polymers saturate at lower values of *H*. Saturation phenomena are observed to be related to the formation of stretched chain-like clusters that percolate the system along the field direction: once polymer chains are almost fully oriented along the field and the system is percolated, no further configurational changes in the magnetic suspension are expected to occur when *H* is increased: for H=(0,1) the field strength is large enough for the differences in $N_{c}$ to disappear. Therefore, entropic contributions to the force and Kelvin forces are expected to also remain constant if *H* is further increased.

The observed evolution of the force profiles with *H* in Figure 3a–d demonstrates that when $\theta =90^{\circ }$ it is possible to tune the number and the position of the stable points together with the range of their attractive basins. This provides a mechanism for selecting different distances of equilibrium between the two large non-magnetic colloidal particles, and bringing them closer or separating them. The general behavior seen in Figure 3a–d can be easily interpreted in a qualitative way using the inverse ferrofluid analogy: when the field is applied perpendicular to the joining axis the situation is akin to having two magnetic dipoles parallel between them but both perpendicular to the axis joining the centers of the two particles. Therefore, according to Equation (Equation 4) a repulsive force among the two induced dipoles emerges. This repulsive contribution will oppose the entropic forces that create an attractive force between the two large colloids, leading to the observed shifts in the force profiles where attractive zones turn into repulsive ones by increasing *H*.

The enlarged oscillations and the more regular patterns in the force profiles observed when *H* is increased are related to the amount of magnetic material able to fit the gap between the two large colloidal particles and how Kelvin forces acting on those particles depend on it. For a qualitative explanation, the best image can be obtained if one analyzes the saturating limit H=(0,1) that in our setup corresponds to colloidal polymer chains being almost straight rods directed perpendicularly to the joining axis (see Figure 3d). In this case, by the symmetry of the problem, Kelvin forces will be non-zero only along the joining axis (x-axis in our setup). These forces are expected to be roughly proportional to $\partial (M_{y}^{2})/\partial x$, where $M_{y}$ is the component of the magnetization along the y-axis. The total Kelvin force acting on each non-magnetic particle can be split into two main contributions: one arising from the magnetic mismatch between the particle and the inner region of the suspension, i.e., the part of the suspension in the gap between the particles, henceforth, we label such a contribution as $F_{K}^{\left(in\right)}$. This contribution always pulls the two particles inwards, i.e., into the gap, therefore, it adds an attractive contribution to the force profile. The second contribution to the Kelvin force emerges from the magnetic mismatch between the particle and the outer or bulk region of the suspension; henceforth, we will label this contribution as $F_{K}^{\left(out\right)}$. This contribution attempts to pull the particle outwards from the gap, i.e., it adds a repulsive contribution to the force profile. For small gaps between the two colloidal particles, $F_{K}^{\left(in\right)}$ is expected to be roughly proportional to ${\langle M_{y}^{2}\rangle }_{gap}/d$, where ${\langle M_{y}^{2}\rangle }_{gap}$ is the averaged magnetization in the gap. On the other hand, $F_{K}^{\left(out\right)}$ is expected to be roughly constant as the bulk magnetization is not expected to change with *d*. If particles are far apart, the magnetization in the region of the gap between the two particles is expected to be similar to that of the bulk, so both contributions are expected to be equal but of opposite sign; therefore, the net total force due to Kelvin forces on each non-magnetic particle is zero. If the gap between particles narrows, then ${\langle M_{y}^{2}\rangle }_{gap}$ will strongly fluctuate with *d*. Thus, for instance, when considering the peak for the force profile in the range d/σ∈(1,2), one should notice that by increasing *d* in this range, the gap widens, but due to steric repulsion among polymers and the repulsion of the polymers with the large colloidal particles, no more magnetic material can enter the gap. Therefore, increasing *d* in this range should lead to a decrease in ${\langle M_{y}^{2}\rangle }_{gap}/d$, i.e., $F_{K}^{\left(in\right)}$ will decay while $F_{K}^{\left(out\right)}$ is roughly constant. Therefore, the net Kelvin force acting on each one of the two non-magnetic colloids will become more repulsive and increase in modulus. Nonetheless, when getting closer to d/σ≈2, there is enough room for a second magnetic polymer chain to fit into the gap and thus $F_{K}^{\left(in\right)}$ becomes in modulus more similar to $F_{K}^{\left(out\right)}$, and the net Kelvin Force reduces. Repulsive peaks in the force profile for larger values of *d* should also follow a similar mechanism. This mechanism explains why the repulsive peaks increase their repulsiveness as the *H* increases (as $M_{y}$ increases with *H*), but it reaches a saturation limit beyond a certain threshold field when polymer chains are almost straight rods. One should bear in mind that in our simulations we use soft potentials for the steric interactions, and that as the gap increases the dependence of ${\langle M_{y}^{2}\rangle }_{gap}$ with *d* becomes more subtle. Furthermore, in addition to the Kelvin forces, there are other contributions to the force profiles related to the way how the magnetic field modifies the polymer conformations. Therefore, one should not expect a regular pattern of stable points at multiples of d/σ, as the exact location of the stable points will depend on the interplay of all previous factors. The reason why for d<σ the force profile is always attractive can also be explained as follows: as no magnetic material can fit into the gap in this case, one expects ${\langle M_{y}^{2}\rangle }_{gap}\approx 0$, and therefore, $F_{K}^{\left(in\right)}\approx 0$. Thus, the net Kelvin force is expected to be repulsive. Nonetheless, at such close distances, as shown in Figure 2, the depletion forces are very attractive. On the other hand, stiffer polymer chains are known to create more attractive depletion interactions than flexible polymers at close distances [130,131]. Thus, when an external field is applied, and the polymers become stiffer, we expect the entropic attractive contribution to be even larger than at H=0. Therefore, the net outcome when *H* is increased is that for d/σ<1 the force profile remains always attractive.

On the other hand, when $\theta =0^{\circ }$, the behavior is far more trivial: Figure 4 shows the force profiles for $N_{c}=\;1,\;10,\;20$ and H=(1,0): the outcome is in all cases an almost linear attractive force profile in which the force strengths decrease with the distance between non-magnetic particles. When comparing Figure 2 with Figure 4, one infers that the effect of an external field applied parallel to the joining axis is to soften attractive forces for short distances, and in turn, promote a considerable enlargement of the range of attraction of the interaction; observe that for d/σ>6, while in Figure 2 the force is almost zero, this is not the case in Figure 4. The analogy with the inverse ferrofluids is also very useful in this case to explain qualitatively what occurs: the situation is akin to having two induced dipoles parallel to the joining axis (in our setup, pointing along the negative x-axis). Therefore, according to Equation (Equation 4), the Kelvin forces for $\theta =0^{\circ }$ add an attractive longe-range contribution to the force profiles. In addition to this contribution, one should also bear in mind the changes to the entropic contribution to the force profiles. This contribution is related to the changes in size and mass distribution of the magnetic polymer chains induced by *H*. Remarkably, for $\theta =0^{\circ }$ in the field saturating limit, see Figure 4, force profiles exhibit values very similar regardless of $N_{c}$. This convergence can be explained by recalling that for large *H*, chains are mostly stretched and orientated along the field forming aggregates that attempt to percolate the system; therefore, in this limit, given the same *R*, ϕ and $\mu^{2}$ one expects the magnetic suspension properties to be quite independent of $N_{c}$.

The inverse ferrofluid analogy can also help us to qualitatively explain what happens when an external field is applied at $\theta \in (0^{\circ },\;90^{\circ })$. Equation (Equation 4) predicts that in those cases, the net effect of Kelvin forces will lead force profiles $f^{\left(x\right)}$ to smoothly change from an attractive to a repulsive profile, as θ is shifted from $\theta =0^{\circ }$ towards $\theta =90^{\circ }$. Figure 5a,b show $f_{2}^{\left(x\right)}$ profiles for H=(0.02,0.02) and H=(1,1), respectively. A comparison of those profiles with those observed in Figure 3a–d and Figure 4 confirms our expectations.

For $\theta \in (0^{\circ },\;90^{\circ })$ and H>0 the force perpendicular to the joining axis ($f^{\left(y\right)}$ in our setup) is non-zero in general. Figure 6a,b plots $f_{2}^{\left(y\right)}$ rather than $f_{2}^{\left(x\right)}$ force profiles for the same cases as in Figure 5a,b. These figures show that for $\theta =45^{\circ }$, independently of the strength of the external field $f_{2}^{\left(y\right)}\geq 0$ for all d/σ. The same results are obtained for $\theta =10^{\circ },80^{\circ }$ (not shown). In our simulations, the largest values of $f_{2}^{\left(y\right)}$ are registered for $\theta =45^{\circ }$. Therefore, excluding the cases $\theta =0^{\circ },\;90^{\circ }$, for any other value of θ, the two non-magnetic colloids exhibit a force contribution out of the joining axis that tends to rotate and align their joining axis with the external field. Thus, the case $\theta =0^{\circ }$ represents a stable point in what refers to the rotation movement of the two non-magnetic particles, while $\theta =90^{\circ }$ represents an unstable point as slight changes in θ will lead to the existence of a pair of forces perpendicular to the joining axis that tend to align both non-magnetic particles with the direction of the external field. It should be observed that the dynamics of the non-magnetic particles are expected to be far from trivial: when particles are close to an angle of $\theta =90^{\circ }$, forces along the joining axis can be either attractive or repulsive, while at the same time, orthogonal forces to the axis push them to align with the external field.

Previous results demonstrate that when a steady external field is applied to a bath of magnetic colloidal polymers, this system can be used as a funneling tool to create linear arrays of non-magnetic particles: the same that occurs with two non-magnetic particles is expected to occur when more non-magnetic particles are present in the interface, as all them will tend to align their joining axis along the direction of the field.

Another point worth remarking is that our simulations show that for values of θ close to $\theta =90^{\circ }$, the component of the force perpendicular to the joining axis increases slowly with the deviation of θ with respect to $\theta =90^{\circ }$. Therefore, if one somehow manages to effectively suppress those small forces perpendicular to the joining axis and stabilize $\theta =90^{\circ }$ against rotations toward the field direction, one could then benefit from the ability to tune via *H* the number and location of the stable points for the force contribution along the joining axis ($f^{\left(x\right)}$ in our setup) to control the distance between the two non-magnetic particles. This stabilization can be accomplished in several ways: either by having a fluid viscous enough or by using an oscillating field rather than a constant one for $\theta =90^{\circ }$ ( e.g., $H=H\;\sin\left(\omega t\right)\hat{y}$ in our setup). In the last case, we benefit from the fact that due to the symmetry of the problem, the component of the force parallel to the joining axis is symmetric with respect to θ, while the component perpendicular to the joining axis is antisymmetric with respect to θ (i.e., for our setup $f^{\left(y\right)}\left(90^{\circ }\right)=-f^{\left(y\right)}\left(-90^{\circ }\right)$ while $f^{\left(x\right)}\left(90^{\circ }\right)=f^{\left(x\right)}\left(-90^{\circ }\right)$). Therefore, if ω is large enough one should be able to obtain, on average, a zero force along the direction perpendicular to the joining axis, $\langle f^{\left(y\right)}\rangle =0$ in our case, while retaining a set of stable points for the contribution of the force parallel to the joining axis. One could envision other forms to stabilize $\theta =90^{\circ }$ against rotations, for instance, making use of rotating fields where the angular rotation of the field is coupled to the rotation of the joining axis, although this can be much more difficult to achieve experimentally.

### 3.3. Dependence of the Force Profiles on ϕ and R/σ

The density of the polymer suspension ϕ, and the size-ratio R/σ between the nonmagnetic colloids of size *R* and the ferromagnetic monomers of size σ forming the polymers are also key parameters to tune the depletion force profiles in these systems. The effect of these parameters on the force profiles is better seen in the field saturating limit. Figure 7 shows the depletion force profiles for $N_{c}=10$ and R/σ=5 for several values of the density ϕ as defined in Equation (Equation 1). For ϕ=0.05 the depletion forces are almost zero which implies that osmotic pressures are very low. As ϕ increases, the strength of the depletion forces acting on the two non-magnetic colloids also increases. One can also infer from that figure that, as the density ϕ increases, the distance at which the change from the attractive to repulsive regime occurs is shifted toward smaller d/σ values: thus, for ϕ=0.15 the crossing from one regime to the other occurs at *d*∼1.6σ, while for ϕ=0.35 it happens circa *d*∼0.6σ. Additionally, as expected, the range of the depletion forces extends towards larger distances as ϕ increases.

In Figure 8 we analyze how the ratio R/σ modifies the force profiles for the case $N_{c}=10$ and density ϕ=0.25 in the field saturating limit. The results show that large values of R/σ imply a stronger attraction force for d<σ while, at the same time, for d>σ the observed repulsive shift largely hampers the existence of stable points within the range of distances we have studied. Based on the analysis of how force profiles depend on the field strength and the dependence of the forces profiles on R/σ, one can propose as a simple rule of thumb that the larger the two non-magnetic colloidal particles are compared to the effective size of the polymers, the smaller the strength of the external field is to obtain a transition from attractive to repulsive depletion force profiles. In turn, this implies that the larger the non-magnetic particles are, the smaller the fields needed to tune the density profiles to exhibit stable points. On the other hand, from the dependence of the force profiles on $N_{c}$ at constant field strength, see for instance Figure 3a–d, one infers that the shorter the colloidal polymers are the smaller the repulsive shift induced in the force profiles at fixed *H*.

## 4. Conclusions

The Langevin Dynamics simulations of a quasi-two-dimensional suspension of ferromagnetic polymers where two large non-magnetic colloids are immersed show that the depletion force profiles are very different when compared to those corresponding to non-magnetic polymers for the same colloidal size ratio. The force profiles in the magnetic case exhibit larger and more regular oscillations that can be related to the variable amount of chains able to fit the gap between the two large colloids depending on the wideness of the gap region. The number of chains in the gap region influences both the osmotic pressure contribution and a new contribution exclusive to the magnetic colloidal polymers due to the emergence of unbalanced Kelvin forces. This imbalance stems from the different magnetic mismatches among ferromagnetic polymers and the two non-magnetic large colloids when one compares the gap region between particles with the region external to the gap. Our results show that larger size ratios or polymer densities lead to enlarged ranges for these forces, and an increase in any of the two parameters increases the repulsive nature of the depletion forces.

Another more practical way to modify the force profiles is related to the introduction of an external magnetic field. By increasing the strength of the magnetic field it is possible to increase the amplitude of the oscillations and offset the force profile towards being more repulsive on average. Suitable field strengths allow for the emergence of new stable points and a certain control of the distances between the two particles at which they occur. Another key factor for the force profiles is the angle θ formed by the field and the axis joining the centers of the two particles. In an external field force, profiles are observed to have a non-zero component perpendicular to the imaginary axis joining the particles. This component leads to a torque onto the two large colloids that tend to align them with the field direction. This lateral component of the force is observed to be at the maximum at $\theta =45^{\circ }$ and has a minimum at $\theta =0^{\circ }$ (stable against rotations) and $\theta =90^{\circ }$ which is unstable as any small perturbation leads the system to evolve towards $\theta \to 0^{\circ }$. At $\theta =0^{\circ }$, force profiles are attractive over the whole range of tested distances; therefore, suspensions of ferromagnetic particles could be used as funneling tools to create aligned clusters of non-magnetic colloids.

The nature of these lateral forces and symmetry arguments (see previous section) suggest that $\theta =90^{\circ }$ orientation can be easily stabilized, for instance by using a fast-time oscillating field always perpendicular to the joining axis. If stabilization is achieved, the dependence of the stable points with the strength *H* of the field can be used in the creation of magnetic colloidal tweezers or magnetic ratchets.

The results obtained in this work envisage the potential of these systems in the field of direct colloidal self-assembly, the fabrication of materials based on self-organized colloidal structures, protein crystallization assistance, shape and size selection of colloidal particles, and the driven motion of colloidal particles and proteins, among other uses.

The present study constitutes a first step toward the understanding of these systems and their potential for different applications. The next steps comprise the study of the depletion force profiles in bulk solutions of magnetic colloidal polymers, as well as the development of analytic frameworks to accurately predict the depletion forces for these systems. We conjecture that for symmetry reasons the largest contribution to the force in a three-dimensional suspension will stem from the slice of polymer chains in the proximity of the geometrical plane that contains the centers of the two large non-magnetic colloids and the field direction. Therefore, although force profiles are expected to be different in quasi-two and three-dimensional suspensions, we do not expect the qualitative behavior to be radically different when comparing both cases.

It should be noted that the quasi-2D system modeled in this work could be, for instance, experimentally realized using air–fluid or fluid–fluid interfaces as already conducted in several works for magnetic particles [132,133,134,135], or using fluid-monolayers [136,137].

We expect our work will stimulate further developments on this subject of increasing scientific interest.

## Figures and Tables

**Figure 1 polymers-16-00820-f001:**
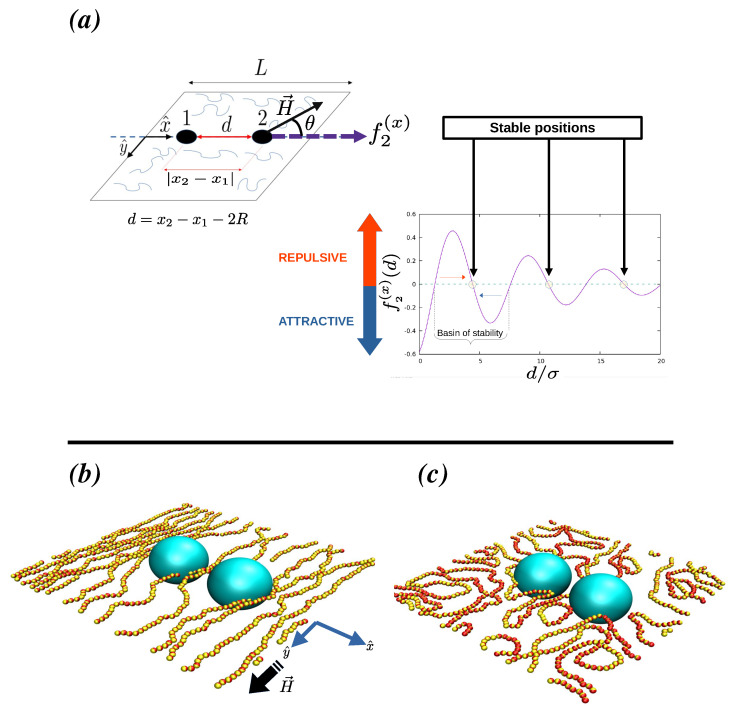
Plot (**a**) shows a schematic diagram showing the simulated system and a typical pair-force profile for the two non-magnetic particles induced by the ferromagnetic suspension of colloidal polymers, in particular, the plot depicts the force profile along the joining axis for the colloidal particle labeled as 2 in the figure. The distance *d* corresponds to the closest distance between the surfaces of the two non-magnetic colloids labeled 1 and 2, i.e., $d=x_{2}-x_{1}-2R$, where *R* is the radius of these particles. We will refer to the direction crossing the center of mass of the two non-magnetic particles as the joining axis: in our setup, this axis is the x-axis. If an external magnetic field (***H***) is applied in a direction lying within the plane, the angle formed by the joining axis and the direction of the field is denoted in this work as θ. Plot (**b**) shows a rendering of the system for the case $R/\sigma =5,\;N_{c}=20,\;\phi =0.25,\;\mu^{2}=5$, for d/σ=2.8 and H=(0,1). Plot (**c**) shows a rendering of the system for the same case as (**b**) but at zero field.

**Figure 2 polymers-16-00820-f002:**
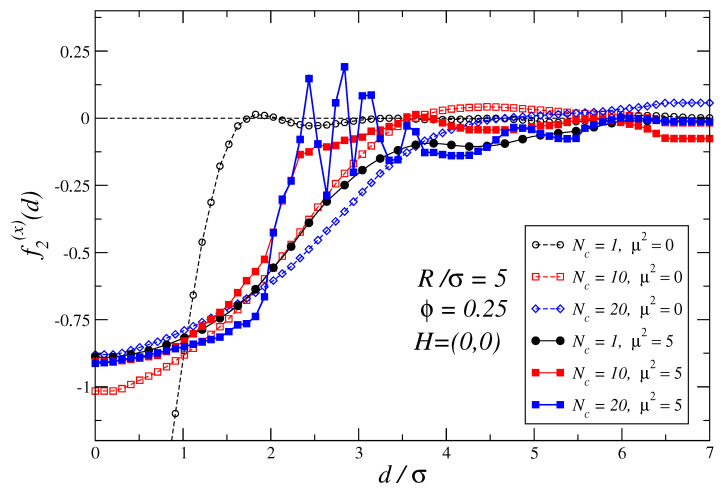
Depletion force profiles are shown for a suspension of non-magnetic, $\mu^{2}=0$, and magnetic, $\mu^{2}=5$, non-linked colloidal particles, $N_{c}=\;1$. In the magnetic case, it corresponds to a simple ferrofluid suspension. The plot also shows the force profiles for suspensions of colloidal polymers of length $N_{c}=\;10,\;20$. All cases plotted in this figure correspond to zero-field. The size ratio of the non-magnetic colloidal particles to monomers is set to R/σ=5. The density of the suspension (see Equation (Equation 1)) is fixed to ϕ=0.25 in all cases. The largest error in the force profiles shown in this figure is ±0.06 at d/σ=2.2 for ($N_{c}=20,\;\mu^{2}=5$) case.

**Figure 3 polymers-16-00820-f003:**
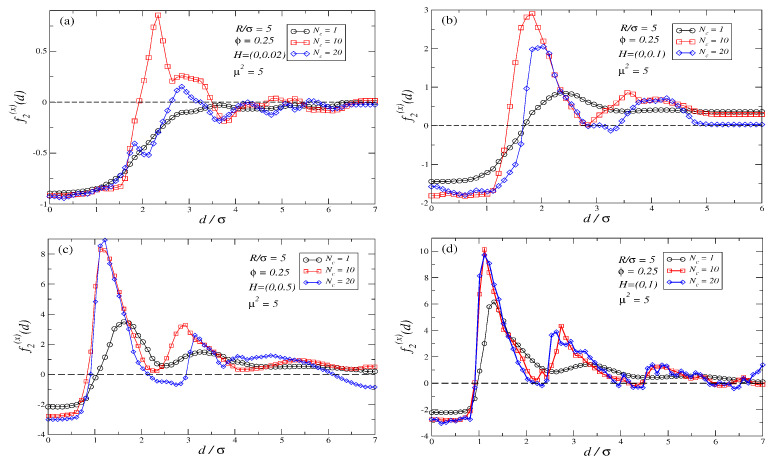
Force profiles for the interaction between the two non-magnetic colloidal particles for suspensions containing different polymer lengths $N_{c}$, for a density ϕ=0.25, and a colloidal size ratio between large and magnetic colloidal particles R/σ=5. Figure (**a**) depicts the case H=(0,0.02), (**b**) corresponds to H=(0,0.1), and figure (**c**,**d**) depicts H=(0,0.5) and H=(0,1.0), respectively. All four cases correspond to magnetic fields applied perpendicular to the joining axis, i.e., $\theta =90^{\circ }$. The largest errors in the force profiles shown in these plots are plot (**a**) ±0.06 at d/σ=1.6 for $N_{c}=20$ case; plot (**b**) ±0.12 at d/σ=1.82 for $N_{c}=20$ case; plot (**c**) ±0.8 at d/σ=1.1 for $N_{c}=20$ case; plot (**d**) ±0.3 at d/σ=1.6 for $N_{c}=20$ case.

**Figure 4 polymers-16-00820-f004:**
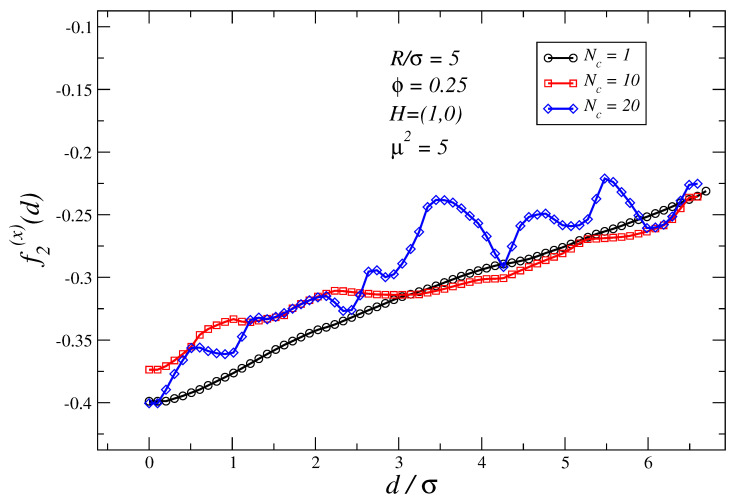
Force profiles for a system with parameters similar to those of Figure 3 but an external magnetic field is applied parallel to the joining axis H=(1,0). The largest error in the force profiles shown in this figure is ±0.013 at d/σ=0.2 for $N_{c}=20$ case.

**Figure 5 polymers-16-00820-f005:**
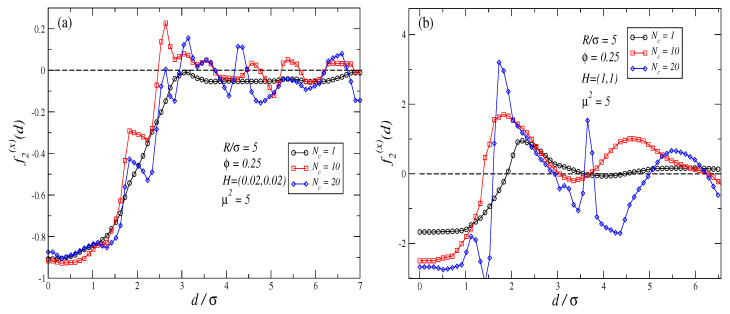
Force profiles corresponding to $f_{2}^{\left(x\right)}$ for the cases (**a**) H=(0.02,0.02), and (**b**) H=(1,1). The largest errors in the force profiles shown in these plots are: plot (**a**) ±0.09 at d/σ=2.64 for $N_{c}=20$ case; plot (**b**) ±0.5 at d/σ=1.62 for $N_{c}=20$ case.

**Figure 6 polymers-16-00820-f006:**
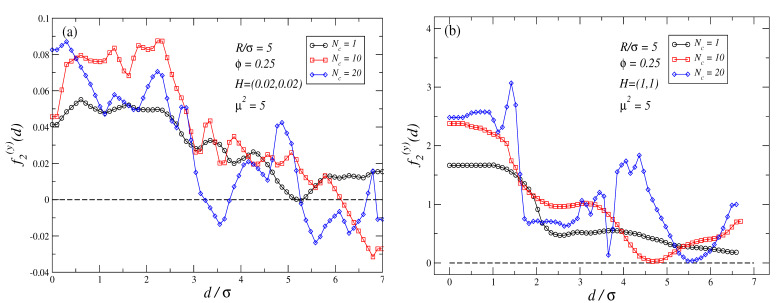
Force profiles corresponding to $f_{2}^{\left(y\right)}$ for the cases (**a**) H=(0.02,0.02), and (**b**) H=(1,1). The largest errors in the force profiles shown in these plots are: plot (**a**) ±0.01 at d/σ=2.54 for $N_{c}=20$ case; plot (**b**) ±0.12 at d/σ=1.42 for $N_{c}=20$ case.

**Figure 7 polymers-16-00820-f007:**
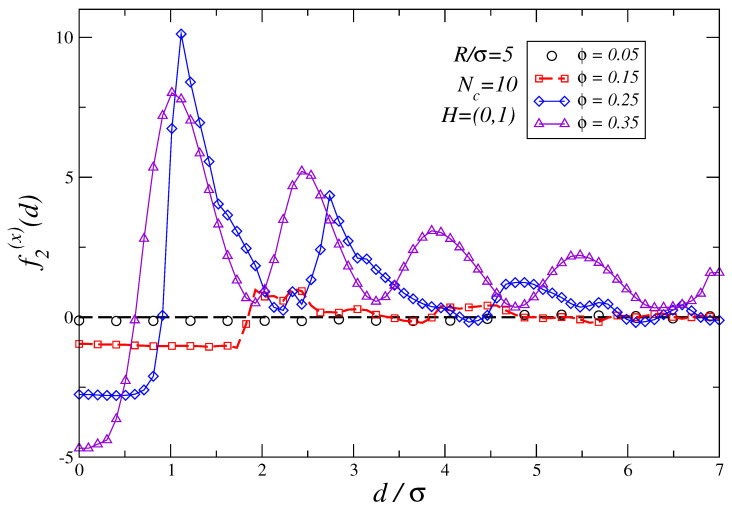
The plot depicts the modifications induced by an increase in the density ϕ of a suspension of colloidal magnetic polymers of size R/σ=5, chain length $N_{c}=10$, when an external field H=(0,1) is applied. The largest error in the force profiles shown in this figure is ±0.3 at d/σ=1.0 for ϕ=0.25 case.

**Figure 8 polymers-16-00820-f008:**
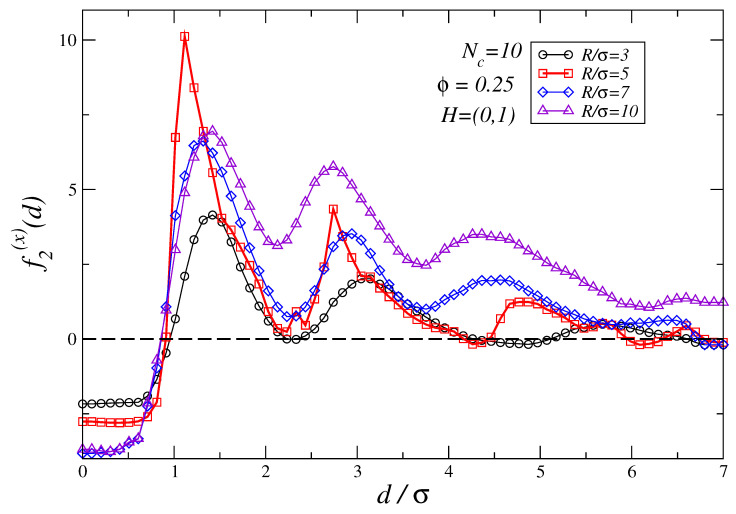
The plot shows how the force profiles for the large non-magnetic colloids vary as a function of the colloidal size ratio R/σ for a solution of colloidal polymers of length $N_{c}=10$ and density ϕ=0.25. An external magnetic field H=(0,1) is applied. The largest error in the force profiles shown in this figure is ±0.3 at d/σ=1.0 for R/σ=5 case.

## Data Availability

Data are contained within the article.

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
