# Peer review of "Depletion Interactions at Interfaces Induced by Ferromagnetic Colloidal Polymers"

_polymers, 2024, doi:10.3390/polym16060820_

Round 1

Reviewer 1 Report

Comments and Suggestions for Authors

Recently, much attention has been paid to liquid and viscoelastic media, the properties (mechanical, electrical, electromagnetic, etc.) of which can be controlled by external influences, including through electric and magnetic fields. Many publications are devoted to materials in which the dielectric (including ferroelectric) medium as a matrix contains magnetic inclusions, and the properties of such a material change due to the transfer of mechanical action from magnetic particles to the matrix continuum as a continuous medium.

In this paper, modeling of the propagation of the effect of a suspension of magnetic colloidal polymers on individual non-magnetic colloidal particles is considered. There is a fairly extensive bibliography devoted to the consideration of interactions taking place in such systems. Since the case of depletion of such interactions has not been theoretically considered before, the problem raised in the manuscript in this aspect is of undoubted interest.

However, in the revised version of the manuscript, an explanation should be given why, when considering the surface density of monomers in a quasi-two-dimensional system (expression 1), normalization does not cover the entire real surface area of L2, but only minus its part occupied by large non-magnetic particles.

However, not all parameters were entered with a sufficiently clear justification. So, the authors, on the one hand, postulate the use of a reduced form of the Boltzmann constant k=1, and on the other hand, introduce the constant ke, normalized to Lennard-Jones potential. Also, for example, one can only guess what the designation r12 means. The choice of the values of magnetic moments used is not fully justified and it is not clear which values they correspond to in real systems. We can only assume that the symbol µ0 indicates the Bohr magneton. However, in line 244, this symbol means the permeability of the vacuum used in the IS unity system. Line 243 contains the expression of magnetic induction B = H + M, which does not correspond to the formulas of any of the systems of units of measurement. If the permeability of the vacuum is mentioned, then the coefficient µ0 should have been present in the formula.

You can also point out some typos, for example, in the phrase “have their center of mass confined in a the same plane.”

Thus, despite the interesting and highly respected approach to modeling the considered system, it is impossible to adequately assess the reliability of the conclusions presented in the article.

The article can be published after the elimination of the noted shortcomings.

Comments on the Quality of English Language

see in the review

Author Response

Reply to referee#1

We would like to thank the referee for the careful reading of the manuscript and for her/his comments and questions. We provide below these lines our answer to the queries of the referee.

(1) - In the revised version of the manuscript, an explanation should be given why, when considering the surface density of monomers in a quasi-two-dimensional system (expression 1), normalization does not cover the entire real surface area of L2, but only minus its part occupied by large non-magnetic particles.

In equation 1, page 2, as the two large colloidal non-magnetic particles interact repulsively with the polymer colloidal chains, we exclude from the effective area available to the polymers the area occupied by the two large colloidal non-magnetic particles. For this reason we subtract to L2 the value of 2 pi R^2. In order to make clear this point in the manuscript we have added the following text (marked in blue in the manuscript) after equation 1 is defined.

In equation 1 we subtract to the total area $L^2$ the area occupied by the two large non-magnetic particles $2 \pi R^2$ in order to get an estimate of the effective area available to the polymer chains which is the observable directly related to the entropy of those chains.”

(2) - not all parameters were entered with a sufficiently clear justification. So, the authors, on the one hand, postulate the use of a reduced form of the Boltzmann constant k=1, and on the other hand, introduce the constant ke, normalized to Lennard- Jones potential. Also, for example, one can only guess what the designation r12 means. The choice of the values of magnetic moments used is not fully justified and it is not clear which values they correspond to in real systems. We can only assume that the symbol µ0 indicates the Bohr magneton. However, in line 244, this symbol means the permeability of the vacuum used in the IS unity system. Line 243 contains the expression of magnetic induction B = H + M, which does not correspond to the formulas of any of the systems of units of measurement. If the permeability of the vacuum is mentioned, then the coefficient µ0 should have been present in the formula

Following the advice of the referee, we have taken different actions in order to improve the description and clarity of the parameters used in the manuscript:

2.1 - Following the definition of T, line 97, we have introduced the following text (marked in blue in the revised manuscript version):

In the previous formula $k_e T_e$ is the product of the Boltzmann constant and the experimental temperature expressed in the same system of units than $\epsilon_e$.”

The expression $T = k_e T_e / epsilon_e$ arises from applying the definition of reduced energy $U = U_e/epsilon_e$ to compute the value of kT in reduced units, making use of the fact we have defined in reduced units the Boltzmann constant as k=1.

2.2- We have modified the caption of the figure 1. We have opted by replacing $r_{12}$ by $x_2-x_1$ in the definition of d in the caption (same definition as given in the text of section 3), in this way we avoid to define a new parameter not necessary.

2.3- In the expression of the modulus of the dipole moment in the reduced unit system , $\mu_{0,e}$ represent the vacuum permeability and \mu_e the value of the dipole. The expression we obtain for the modulus in the reduced system arises from considering the energy of two magnetic monomers at close contact with their dipoles aligned nose-to-tail in the reduced units system using equation 4, i.e. $-2 \mu^2 / \sigma^3$ and equating the previous expression to the reduced energy arising from dividing the experimental magnetic energy of that two-dipole system by \epsilon_e, the experimental well depth of the interaction between two monomers, which in electromagnetic SI units is equal to $-2/(4 \pi \mu_{0,e}) \mu_e^2 / \sigma_e^2$ as the distance between the two monomers at close contact is $\sigma_e$. It should be noticed that in reduced units we take $\sigma=1$, therefore we are equating $ -2 \mu^2 = - 2/(4 \pi \mu_{0,e}) \mu_e^2 / \sigma_e^2$, and from here it is derived the expression of the modulus of the dipole moment in the reduced unit system.

In order to improve the manuscript and avoid any possible confusion we have replaced

Two monomers are assumed to interact pair-wise as point dipoles according to the potential,”

by the following text

“… where $\mu_{o,e}$ is the vacuum permeability, and SI electromagnetic units are assumed to express the experimental magnetic energy associated to a system of two magnetic monomers at close contact with their dipoles aligned nose-to-tail. In our reduced unit system two monomers are assumed to interact pair-wise as point dipoles according to the potential, ...”

2. 4 - We have corrected the expression of the magnetic induction $B = H + M$ given in former line 243 by $B = \mu_0 H + M$. In this way SI electromagnetic units are always used along the manuscript.

(3) You can also point out some typos, for example, in the phrase “have their center of mass confined in a the same plane.”

We have checked thoroughly the manuscript for typos and corrected them. We thank again the referee for the thoroughly revision of the manuscript.

Reviewer 2 Report

Comments and Suggestions for Authors

The paper “Depletion Interactions at Interfaces Induced by ferromagnetic colloidal polymers” examines the contribution of a ferromagnetic polymer suspension to the depletion interaction between two non-magnetic colloidal particles. The article is well written and contains a detailed list of depletion interaction studies in the introduction. The obtained calculation results open up the possibility of controlling the depletion interaction between colloidal particles by changing the number of monomers in the polymer molecule, the strength and direction of the applied magnetic field. I believe that the article will stimulate experimental research in this area. I have a few comments to the authors:

1) The insert |x2-x1|in Figure 1 is compressed in the horizontal direction. Please fix this.

2) Do you have a qualitative explanation for the oscillations in the range (2;4) of the Nc=20 μ^2=5 line in Fig. 2? For other d/σ ratios the situation looks quite smooth.

 3) Extension to a 3D model is essential since the idealized 2D model proposed here is unlikely to be realized experimentally.

Author Response

Reply to referee#2

We would like to thank the referee for her/his comments and questions. We provide below these lines our reply to them

(1) The insert |x2-x1|in Figure 1 is compressed in the horizontal direction. Please fix this.

We have fixed the figure 1 accordingly.

(2) Do you have a qualitative explanation for the oscillations in the range (2;4) of the Nc=20 μ^2=5 line in Fig. 2? For other d/σ ratios the situation looks quite smooth.

We thank again the referee for raising this very interesting question. In order to answer the question we have included the following explanation in the section 3.1 of the manuscript circa line 226, (we have highlighted in blue the new introduced text in the manuscript):

The oscillations with a larger amplitude and shorter wavelength observed in figure \ref{f2} for ferromagnetic polymers in the range $d/\sigma \in (2,4)$ can be related to the fact than ferromagnetic polymers can pack more densely than their non-magnetic polymer counterparts in the region gap between the two large non-magnetic colloids. In the case of the ferromagnetic polymers, their dipoles tend to orientate in a nose-to-tail configuration which leads in average to more stretched polymer conformations. For short ferromagnetic chains, the local persistence length is expected to increase with $N_c$. As ferromagnetic colloids can therefore pack more densely, the required change in the gap distance $\Delta d$ that determines if another polymer can be easily accommodated or not into the gap shortens. This is expected to lead to oscillations of shorter wavelength as the length of the oscillation in the depletion force is expected to be influenced by $\Delta d$. In turn, larger amplitude oscillations in the depletion force profiles are expected to occur as the density gradients created by a slightly varying distance $d$ will be larger. It should be mentioned that oscillations in the osmotic pressure with wall separation related to the density has been also observed in systems of colloidal particles immersed in a sea of charged nanoparticles\cite{2009-fazelabdolabadi}.”

(3) Extension to a 3D model is essential since the idealized 2D model proposed here is unlikely to be realized experimentally.

The quasi-2D model could be physically realized by using an interface air-fluid or fluid-fluid where the density of the fluid (water/oil/Octane) is matched with that of the large colloidal particle and the ferromagnetic monomers so polymers and particles are located at the interface due to their buoyancy. In order to match density of particles with the fluid, one could resort to do colloidal particles with a cobalt or magnetite shell/core with a core/shell of latex or other low-density polymeric substances. One can find in literature studies of magnetic colloidal particles at an interface, See for instance

- F. Martínez-Pedrero, “Static and dynamic behavior of magnetic particles at fluid interfaces”, Adv. In Colloid and Interface Science, 284, 102233, (2020). https://doi.org/10.1016/j.cis.2020.102233

- F. Martínez-Pedrero et al , “ Static and dynamic self-assembly of pearl-like chains of magnetic colloids confined at fluid interfaces”, Small, 17, 2101188, (2021). DOI: 10.1002/smll.202101188

- N. Vandewalle et al, “Mesoscale structures from magnetocapillary self-assembly”, The European Physics Journal, E, Soft Matter, 36, 9941, (2013). doi: 10.1140/epje/i2013-13127-7

- Cappelli, S. (2016). “Magnetic particles at fluid-fluid interfaces : microrheology, interaction and wetting.” [Phd Thesis 1 (Research TU/e / Graduation TU/e), Biomedical Engineering]. Technische Universiteit Eindhoven

- U.B.Gunatilake et al, “ Magneto Twister: Magneto Deformation of the Water–Air Interface by a Superhydrophobic Magnetic Nanoparticle Layer”, Langmuir, 38, 3360, (2022). [https://doi.org/10.1021/acs.langmuir.1c02925]

Also, if the difference in size between the non-magnetic colloidal particles and the ferromagnetic polymers is not very large, experiments could be conducted on colloidal monolayers where particles are for instance suspended in a thin layer of ferrofluid or water subphase, see for instance:

- Yang, et al. Soft Matter, 11, 2404, (2015). DOI: https://doi.org/10.1039/C5SM00009B

- Lefebure et al, J. Phys. Chem. B, 102, 15, 2733 (1998). DOI: https://doi.org/10.1021/jp980403+

Therefore, due to all previous reasons mentioned above we do not share the opinion of the referee about the fact that the 2D model is unlikely to be realized. Colloidal science has made in recent years quantum leap advances that should make endurable the study of quasi 2D systems. Our opinion is that this quasi-2D systems are an step forward towards the ulterior study of 3D-systems where it can be sometimes more difficult to perform accurate experiments.

Though it would certainly be very interesting to compare the force profiles for a quasi-2D system with those obtained in 3D, the computational study of 3D systems is out of our computational capabilities. The reason of it is that to avoid unphysical effects periodic boundary conditions must be used, and in order to avoid forces to be influenced by the presence of neighbouring replica cells, large simulation cells must be used with only two large colloidal particles in it. For a quasi-2D system we use 782 magnetic particles for a squared simulation cell when we set the size to 50 x 50. Due to the long time needed to properly average forces acting on the two large non-magnetic colloids, the simulation of the force for a given distance d between the two non-magnetic colloidal particles takes circa 24h using 4 cpus running in parallel in a modern one year old workstation. The simulation of an analogous 3D system will imply circa 50000 particles and take about 1500 days using the same methodology. It should be noticed than in these case most of the computational effort will be put on moving the sea of magnetic colloidal polymers located far from the plane of interactions, being that last one, due to the above mentioned symmetries, the real zone of interest. Shortening the cell simulation along the direction perpendicular to the plane containing the dipoles is risky as one can introduce unphysical effects due to the proximity of large colloidal particles between the simulation cell and the upper/bottom replica cells that would be at short distance.

In order to reinforce in the manuscript the idea that quasi-2D systems are realizable and can be a very useful step towards the understanding of the forces in these magnetic systems, we have added at the end of the conclusions section the following text (highlighted in blue in the manuscript):

The present study constitutes a first step towards the understanding of these systems and their potential for different applications. Next steps comprise the study of the depletion force profiles in bulk solutions of magnetic colloidal polymers, as well as the development of analytic frameworks to accurately predict the depletion forces for these systems. We conjecture that for symmetry reasons the largest contribution to the force in a three-dimensional suspension will stem from the slice of polymer chains in the proximity of the geometrical plane that contains the centres of the two large non-magnetic colloids and the field direction. Therefore, although force profiles are expected to be different in quasi-two and three dimensional suspensions, we do not expect the qualitative behaviour to be radically different when comparing both cases.

It should be noted that the quasi-2D system modelled in this work could be for instance experimentally realized using air-fluid or fluid-fluid interfaces as already done in several works for magnetic particles\cite{2020-martinez-pedrero,2021-martinez-pedrero,2013-vandewalle,2022-gunatilake}, or using fluid-monolayers \cite{2015-yang, 1998-lefebure}.”

Reviewer 3 Report

Comments and Suggestions for Authors

Authors have performed Langevin dynamics simulations to study the pair-depletion interactions between two non-magnetic soft colloidal particles surrounded by a suspension of ferromagnetic colloidal polymers confined in a quasi-two dimensional geometry. The found that the natural behaviour of the system is to exert forces on the two non-magnetic particles that tend to rotate the particles and align their joining axis with the direction of the external field, which suggests that magnetic suspension of colloidal polymers could be used as a funnelling tool to create linear arrays of non-magnetic particles. These results might provide new insights into the creation of magnetic colloidal tweezers and ratchets.

Basically, this work allows useful information and is well written. I might recommend its publication after addressing the following points:

1) The Langevin dynamics is used in the simulations, which however cannot consider the potential hydrodynamic effect. So, how does this effect impact the results?

2) There is only line plots in the results. Can some structural images be provided to present a more definite relation between the spheres?

3) The current work consider the 2D cases. How about 3D case? The authors should at least provide some brief discussions regarding this aspect.

4) The fluctuation of the force curves in each figure seems too large. How many runs have been performed for the results? If possible, error bar should be included.

5) The following works might be included for an enhanced background.

Chem. Soc. Rev. 2023, 52, 6806–6837; Phy. Rev. Lett. 2023, 131, 134002

Comments on the Quality of English Language

Minor editing of English language required.

Author Response

Reply to referee#3

We would like to thank the referee for her/his comments and questions. We provide below these lines our reply to them

1) The Langevin dynamics is used in the simulations, which however cannot consider the potential hydrodynamic effect. So, how does this effect impact the results?

We thank the referee for raising this very interesting question about the effects of hydrodynamics. In order to provide further information in the manuscript about the expected effect of hydrodynamics we have included at the end of the section 2 where the numerical method is explained the following text (highlighted in blue in the manuscript)

We expect our simulations to give the correct equilibrium properties in the canonical ensemble in spite of the fact that no hydrodynamic effects have been included in the model. This claim is based on the fact that as far as the generalized equation of Smoluchowski with hydrodynamics (GESH) can be used as a valid description of the system, its stationary solution is the density probability function of the canonical ensemble\cite{2006-37thIFFJulich}. The GESH equation is expected to hold on Brownian time and length scales when both particles and solvent are quasi-inertia free. That amounts to say that in the scale we are performing our measurements the inertial forces must be very small compared to the forces arising from the total potential, the Brownian forces, and the forces of friction with the solvent. We assume that in our system previous conditions hold.”

2) There is only line plots in the results. Can some structural images be provided to present a more definite relation between the spheres?

We have introduced a figure 1b and 1c where a rendering of the system for the case $R/\sigma=5, ~N_c=20, ~\phi=0.25, ~\mu^2=5$, for $d/\sigma=2.8$ and $\bH=(0,1)$ and the same system but at zero field are now shown. We hope these added plots will provide a good insight about the relation between spheres and how the magnetic polymer chains respond to a large external field, and how do they distribute at zero field.

3) The current work consider the 2D cases. How about 3D case? The authors should at least provide some brief discussions regarding this aspect.

We thank the referee for raising this very interesting aspect. We have introduced the following text (highlighted in blue in the manuscript) at the end of our conclusions to comment about the 3D modelling and also reply the very similar question raised by referee#2:

The present study constitutes a first step towards the understanding of these systems and their potential for different applications. Next steps comprise the study of the depletion force profiles in bulk solutions of magnetic colloidal polymers, as well as the development of analytic frameworks to accurately predict the depletion forces for these systems. We conjecture that for symmetry reasons the largest contribution to the force in a three-dimensional suspension will stem from the slice of polymer chains in the proximity of the geometrical plane that contains the centres of the two large non-magnetic colloids and the field direction. Therefore, although force profiles are expected to be different in quasi-two and three dimensional suspensions, we do not expect the qualitative behaviour to be radically different when comparing both cases.

It should be noted that the quasi-2D system modelled in this work could be for instance experimentally realized using air-fluid or fluid-fluid interfaces as already done in several works for magnetic particles\cite{2020-martinez-pedrero,2021-martinez-pedrero,2013-vandewalle,2022-gunatilake}, or using fluid-monolayers \cite{2015-yang, 1998-lefebure}.”

As an extra comment we would like to explain to the referee that though it would certainly be very interesting to compare the force profiles for a quasi-2D system with those obtained in 3D, the computational study of 3D systems is out of our computational capabilities. The reason of it is that to avoid unphysical effects periodic boundary conditions must be used, and in order to avoid forces to be influenced by the presence of neighbouring replica cells, large simulation cells must be used with only two large colloidal particles in it. For a quasi-2D system we use 782 magnetic particles for a squared simulation cell when we set the size to 50 x 50. Due to the long time needed to properly average forces acting on the two large non-magnetic colloids, the simulation of the force for a given distance d between the two non-magnetic colloidal particles takes circa 24h using 4 cpus running in parallel in a modern one year old workstation. The simulation of an analogous 3D system will imply circa 50000 particles and take about 1500 days using the same methodology. It should be noticed than in these case most of the computational effort will be put on moving the sea of magnetic colloidal polymers located far from the plane of interactions, being that last one, due to the above mentioned symmetries, the real zone of interest. Shortening the cell simulation along the direction perpendicular to the plane containing the dipoles is risky as one can introduce unphysical effects due to the proximity of large colloidal particles between the simulation cell and the upper/bottom replica cells that would be at short distance.

4) The fluctuation of the force curves in each figure seems too large. How many runs have been performed for the results? If possible, error bar should be included.

We have run five simulations using a different initial random seed for each point shown in the plots. In order to make this point clear in the manuscript we have added the following text (highlighted in blue in the manuscript) towards the end of section 2 at the end of the next to last paragraph of that section:

Five runs using different initial random seeds have been performed in order to obtain the averaged force profiles shown in this work.”

The reason of not having performed more runs to obtain the averages stems from the limitations imposed by our computational capabilities: for a typical simulation in our study we use 782 magnetic particles for a squared simulation cell of size 50x50. Due to the long time needed to properly average forces acting on the two large non-magnetic colloids, the simulation of the force for a given distance d between the two non-magnetic colloidal particles takes circa 24h using 4 cpus running in parallel in a modern one year old workstation.

In reference to providing an estimate of the errors in our measurements, as many of the force profiles are quite close ones to the others, adding a error bar to each point of each force profile we consider could make quite difficult to visualize correctly the force profiles and the trends displayed by them. For this reason, in order to provide a upper threshold to the error in our measurements we have decided to provide in the caption of each figure the largest error for each plot there represented. These values have been highlighted in blue in the captions of the figures in the manuscript.

5) The following works might be included for an enhanced background. Chem. Soc. Rev. 2023, 52, 6806–6837; Phy. Rev. Lett. 2023, 131, 134002

We thank the referee for pointing out these very enlightening and recent references. We have added them in the introduction section of the manuscript.

Reviewer 4 Report

Comments and Suggestions for Authors

In principle, the topic of the article “Depletion interactions at interfaces induced by ferromagnetic colloidal polymers” is suitable for the journal “Polymers, which provide a detailed mechanistic study via Langevin numerical simulations to the depletion pair-interaction force profiles for two non-magnetic colloidal particles immersed in a suspension of magnetic colloidal polymers. The author has conclusively discussed all the results. However, the author is encouraged to justify the following observations noted.

1.     Abstract is shallow descriptive. The authors are suggested to concise it accordingly and make it meaningful.

2.     In the introduction section: extensive and appropriate literature is requested to explore and cite numerous characterization techniques used for the depletion-interactions of a pair of non-magnetic colloidal particles.

3.     Although there is a good collection of literature but with very less explanation and story line, there is requirement of proper writing approach.

4.     The Conclusion section needs improvement. It should be brief and conclusive. The authors have added some major discussion parts in conclusion. Kindly rewrite it and make sure it is meaningful.

Comments on the Quality of English Language

Should be improved. Abstract and conclusion are shallow descriptive.

Author Response

Reply to referee#4

We would like to thank the referee for her/his comments and questions. We provide below these lines our reply to them

1. Abstract is shallow descriptive. The authors are suggested to concise it accordingly and make it meaningful.

We have rewritten the whole abstract making it more meaningful and concise way following the advice of the referee.

2 & 3. In the introduction section: extensive and appropriate literature is requested to explore and cite numerous characterization techniques used for the depletion-interactions of a pair of non-magnetic colloidal particles. Although there is a good collection of literature but with very less explanation and story line, there is requirement of proper writing approach.

Following the advice of the referee we have rewritten part of the introduction and added several references in order to do a proper writing approach and explain the different characterization techniques used in the study of the depletion-interactions in the case of non-magnetic colloids.

4. The Conclusion section needs improvement. It should be brief and conclusive. The authors have added some major discussion parts in conclusion. Kindly rewrite it and make sure it is meaningful.

We have modified the conclusion section following the advice of the referee to make it more brief, conclusive and meaningful.

Round 2

Reviewer 1 Report

Comments and Suggestions for Authors

The authors have significantly improved the manuscript in accordance with the numerous wishes of the reviewers and mainly took into account the comments made, so the manuscript can be published in the submitted form

Reviewer 3 Report

Comments and Suggestions for Authors

I recommend its publication as is.